# Synergistic Effects of Gibberellic Acid, Forchlorfenuron, Thidiazuron, and Brassinosteroid Combinations on Seedless Berry Development and Quality Enhancement in ‘Shine Muscat’ and ‘Red Muscat of Alexandria’ Grapes

**DOI:** 10.3390/biology14091270

**Published:** 2025-09-15

**Authors:** Pengcheng Yang, Zishu Wu, Boyang Liu, Lei Wang, Shiping Wang

**Affiliations:** School of Agriculture and Biology, Shanghai Jiao Tong University, Shanghai 200240, China; jasonyoung@sjtu.edu.cn (P.Y.); 15622721244@163.com (Z.W.); lby020150910025@sjtu.edu.cn (B.L.)

**Keywords:** plant growth regulators, berry development, gibberellic acid, forchlorfenuron, thidiazuron, streptomycin, TSS, rachis diameter, pedicel structures, Shine Muscat, Red Muscat of Alexandria

## Abstract

Market analysis indicates that large, seedless grapes with good flavor are highly popular. To investigate this, we evaluated the effects of five plant growth regulators—gibberellic acid (GA), 24-epibrassinolide (EBR), forchlorfenuron (CPPU), thidiazuron (TDZ), and streptomycin (SM)—on two grape varieties ‘Shine Muscat’ and ‘Red Muscat of Alexandria’. Regulators were applied after anthesis, and berry size, sweetness, and microscopic features were assessed as the fruit matured. Combining gibberellic acid with either thidiazuron or forchlorfenuron significantly enhanced seedlessness and produced substantially larger berries. Gibberellic acid application also thickened fruit stalks and delayed sugar accumulation. Conversely, higher doses of 24-epibrassinolide and streptomycin inhibited growth. Microscopic analysis revealed distinct mechanisms: thidiazuron and forchlorfenuron primarily enlarged the outer flesh (mesocarp) and skin (exocarp), potentially strengthening the peel, while gibberellic acid promoted the enlargement of internal vascular tissues. Importantly, the two grape varieties exhibited differential responses to the treatments, highlighting that a universal application strategy is not feasible. These findings demonstrate that selecting appropriate regulator combinations and dosages can effectively produce larger, seedless grapes while maintaining excellent eating quality. They also provide growers with practical guidance on application timing and methodology.

## 1. Introduction

Grapevine (*Vitis vinifera* L.) ranks among the most important fruit commodities worldwide, with steadily growing economic value and market demand [1]. Modern consumers increasingly demand seedless, large-berried, flavor-rich fruit with strong storability and transport tolerance as the market mainstream. To meet these demands, the application of exogenous plant growth regulators (PGRs) has become an indispensable, key technology in modern viticulture [2,3,4].

Among them, the combined use of gibberellic acid (GA_3_) and cytokinins—particularly forchlorfenuron (CPPU)—has been widely adopted for inducing parthenocarpy and promoting berry enlargement, with well-documented effectiveness [2,3,5,6]. For example, in ‘Wuhe Cuibao’ grapes, different combinations of GA_3_ and CPPU enhanced berry enlargement and influenced fruit quality [2]. Similarly, in ‘Early Sweet’ grapes, GA_3_ and CPPU improved yield and berry development [3]. The GA_3_ + CPPU combination has also been shown to improve the light response curve and fruit quality in ‘Shine Muscat’ [7]. However, the limitations of this classic regimen are increasingly evident: high doses or improper use often cause a series of adverse effects, such as excessive lignification and thickening of the pedicel and rachis, thicker pericarp, coarse mouthfeel, impeded sugar accumulation, and delayed ripening [8,9]. These quality deteriorations significantly undermine market value and run counter to the goal of producing premium fruit. Therefore, optimizing PGR strategies and exploring new, efficient PGR combinations with fewer side effects—so as to maintain berry enlargement and parthenocarpy while comprehensively improving internal and external quality—has become a major challenge for industry upgrading and a frontier hotspot in grape research [10].

To address these challenges, this study aims to systematically examine the synergistic potential of multiple PGRs. GA_3_, a key hormone promoting cell elongation, used together with cytokinins (such as CPPU or thidiazuron, TDZ) that promote cell division, generates synergistic effects on parthenocarpy and berry enlargement—this is the current technological foundation [7,11]. However, precisely because of this strong growth stimulation, the intrinsic balance of fruit development can be perturbed, leading to quality issues.

To overcome this bottleneck, we introduced 24-epibrassinolide (EBR). As a brassinosteroid (BR)-type regulator, EBR regulates plant growth, development, and stress responses [12,13]. Although direct studies on the combined effects of EBR with GA_3_ or CPPU on grape quality remain relatively limited, indirect evidence and related research support the potential of EBR to optimize fruit quality. For instance, BRs improve fruit parameters such as cluster weight, berry weight, and berry volume [13]. In the seedless grape ‘Tas-A-Ganesh’, applying brassinolide and benzyladenine (individually or in combination) at 7 or 15 days after fruit set, or as two dips at both time points, enhanced cluster and berry traits (including cluster weight, berry weight, berry volume, and berries per cluster) [13]. Studies show that EBR acts synergistically with jasmonic acid (JA) [14] and abscisic acid (ABA) [15], but in grapes, investigations into EBR’s synergy with other PGRs are still in early stages, and systematic assessments of the optimal EBR concentrations for quality improvement are scarce.

To explore new pathways for inducing parthenocarpy, we incorporated streptomycin (SM). Although SM is not a traditional plant hormone, it specifically inhibits normal ovule development to induce parthenocarpy [16]. Studies indicate that applying a combination of streptomycin and gibberellin 1–2 weeks before anthesis significantly increases the parthenocarpic rate in cultivars such as ‘Muscat Bailey A’ [17]. However, whether SM acts synergistically or as a substitute when combined with EBR or cytokinins (TDZ/CPPU) for parthenocarpy in grapes remains largely unexplored. This could not only provide a new, efficient route to parthenocarpy but also reduce reliance on high-dose GA_3_, thereby mitigating some of its side effects at the source. As an antibiotic, SM faces substantial restrictions and risks in agricultural use: its application promotes the emergence and dissemination of antibiotic resistance genes (ARGs) in environmental microbes, which may spread via horizontal gene transfer to human pathogens and pose public health risks [18]. The European Union imposes strict controls (under Regulation (EC) No 1107/2009) [15], and SM is not approved for routine use as a plant protection product. Per Regulation (EC) No 396/2005 [16], when a specific maximum residue limit (MRL) is not established, the default MRL is 0.01 mg/kg—difficult to satisfy in practice. When employing SM to open new avenues for parthenocarpy, residual and resistance risks must be systematically evaluated.

In the choice of cytokinins, this study conducted a parallel comparison between traditional CPPU and the more active thidiazuron (TDZ). As a phenylurea-type cytokinin, TDZ possesses substantially higher activity than CPPU [17]. A study on ‘Shine Muscat’ revealed that both TDZ and CPPU reduce pollen germination after treatment and cause morphological abnormalities in stamens and pollen grains, suggesting that these two regulators affect grape reproductive development via similar routes [14]. Therefore, investigating TDZ’s potential in berry enlargement, identifying its optimal application concentrations to achieve or surpass CPPU’s effects, and concurrently reducing negative impacts, constitute a current research focus.

The effectiveness of PGR treatments varies significantly among different grape cultivars, highlighting the importance of cultivar-specific optimization strategies [19,20,21]. *Vitis vinifera* cultivars demonstrate superior sensitivity to GA_3_ and other growth-promoting regulators due to their enhanced auxin–gibberellin synergistic pathways and more efficient sugar transport capabilities [22]. In contrast, *Vitis labruscana* × *Vitis vinifera* hybrids show heightened sensitivity to ABA-type regulators, attributed to specific promoter structures and elevated expression of stress-response genes [23]. These genetic and physiological differences require experiments on different varieties to determine the concentration and combination of plant growth regulators most suitable for a particular variety.

Against this background, we articulate testable hypotheses: (i) the joint application of GA_3_ with cytokinins (CPPU or TDZ) will act synergistically rather than additively on parthenocarpy and berry enlargement, yielding higher seedless rates and stronger increases in berry size and quality traits; (ii) building on prior studies showing that EBR at concentrations below 1 mg L^−1^ can promote berry quality, yet with the optimal dose insufficiently explored, we hypothesize that, within this sub-milligram range, the quality-enhancing effect becomes more pronounced as concentration decreases; and (iii) pre-anthesis application of streptomycin (SM) can serve as an alternative or augmenting inducer of parthenocarpy that reduces GA_3_ demand, although its negative effect on berry size requires mitigation through interactions with TDZ/CPPU. To test these hypotheses, we designed a factorial scheme covering 12 combinations of GA_3_, CPPU, TDZ, EBR, and SM in two cultivars (‘Shine Muscat’ and ‘Red Muscat of Alexandria’), applied at two developmental windows—full bloom (D0) and 14 days after anthesis (D14)—and integrated morphological, physicochemical, and histological readouts to resolve main and interaction effects. Relative to existing work, our study is innovative in its cross-hormone, time-staged factorial design; in tightly coupling whole-berry phenotypes with pedicel and pericarp histomorphometry to provide mechanistic insight; and in deriving cultivar-specific optimization strategies. In brief, we show that in ‘Shine Muscat’, GA_3_ + TDZ or GA_3_ + CPPU approached 100% seedlessness; TDZ/CPPU markedly increased berry mass and diameter while thickening pericarp epidermis and pedicel cortex; GA_3_ increased mass but promoted rachis thickening and reduced TSS; EBR at ≥0.2 mg L^−1^ inhibited growth; and SM decreased berry size, which was partially mitigated by TDZ. By contrast, ‘Red Muscat of Alexandria’ exhibited generally weaker responses, underscoring genetic constraints on PGR responsiveness. These outcomes support a practical framework that prioritizes TDZ/CPPU as core agents for enlargement, applies GA_3_ with careful dose and timing control, explores low-dose EBR primarily for quality modulation, and restricts SM to research or strictly regulated contexts with concurrent assessment of residue and resistance risks.

## 2. Materials and Methods

### 2.1. Plant Material and Growing Conditions

Uniform, healthy vines of two table-grape cultivars —Shine Muscat (20 vines) and Red Muscat of Alexandria (70 vines)—were cultivated under identical controlled greenhouse conditions at Shanghai Jiao Tong University Viticulture Greenhouse (121°29′ E, 31°11′ N, Shanghai, China), with standard ampelotehchnical measures.

The experiment was conducted from March to August 2023 in Shanghai under the humid subtropical monsoon climate regime. During this experimental period, the East Asian monsoon system dominated the regional weather patterns, resulting in a gradual environmental transition from mild and humid spring conditions to hot and humid summer conditions. As the season progressed, daylength increased progressively with relatively abundant solar radiation, accompanied by intermittent precipitation events. Standard greenhouse ventilation and shading management practices were implemented to mitigate temperature and humidity fluctuations. According to meteorological data from the Shanghai Meteorological Bureau (http://sh.cma.gov.cn/ (accessed on 15 April 2024)), the average temperature during March–May was 20.4 °C with relative humidity ranging from 35% to 85%, while June–August recorded an average temperature of 30.9 °C with relative humidity ranging from 70% to 95%.

Both cultivars were cultivated in the same greenhouse under identical management, including regular irrigation (twice weekly), balanced fertilization (water-soluble fertilizer applied once a week before anthesis, reduced to once every two weeks after anthesis), canopy management (shoot positioning and leaf removal), and pest control measures.

Red Muscat of Alexandria: This cultivar is a red bud mutation of the Eurasian species *Vitis vinifera* ‘Muscat of Alexandria’, originally from Egypt [24]. The variety was introduced to Shanghai Jiao Tong University from Okayama University, Japan in 1998, where it was grafted in the university farm orchard and first fruited in 2001 [24]. The fruit skin color changed from the green of the parent variety to bright red, with this red coloration being stably inherited through multiple propagations. The variety was officially recognized and named by the Shanghai Municipal Crop Variety Approval Committee in 2006. Clusters weigh over 1 kg, with elliptical berries weighing 6–7 g each. The fruit skin is moderately thick and appears pink when covered with bloom. The flesh is crisp and juicy, with soluble solids content of 16–21%, titratable acidity of 0.3–0.4%, and juice yield of 85%. Each berry contains 2–3 seeds [24,25].

Shine Muscat: This is a diploid table grape variety developed through hybridization of ‘Akitsu-21’ and ‘Hakunan’ at the National Institute of Fruit Tree Science in Japan [26]. The variety is characterized by large berries with yellow-green skin, crisp and juicy flesh with muscat flavor, high soluble solids content, and low acidity [26]. Clusters are conical in shape, weighing 600–800 g [27]. Berries are oval-shaped, weighing 8–12 g each [28], with thin skin that turns golden yellow at maturity [26]. The flesh is crisp and refreshing with abundant juice, soluble solids content of 18–22% [28,29], and titratable acidity of 0.5–0.6% [27]. The seedless trait is stable [28]. The maturity period occurs 20–22 weeks after flowering, when the sugar–acid ratio is optimal [28,29].

### 2.2. Experimental Design

For ‘Shine Muscat’, a total of 230 clusters were counted and evenly allocated across 12 treatment groups (approximately 19 clusters per treatment, serving as biological replicates), with each group comprising clusters originating from at least three different vines. During cluster thinning, each cluster retained at least 50 berries. For ‘Red Muscat of Alexandria’, a total of 135 clusters were counted and evenly allocated across 12 groups (approximately 11 clusters per treatment, serving as biological replicates). Because each vine typically bore 2–3 clusters, the evenly allocated clusters necessarily originated from ≥3 vines per group, ensuring sufficient biological replication. During cluster thinning, each cluster was maintained with at least 50 berries.

To minimize environmental bias, clusters were randomly assigned to treatment groups using a systematic randomization approach within the greenhouse layout. This ensured that each treatment group contained clusters from different spatial locations within the greenhouse, reducing potential confounding effects from microclimatic variations.

Full bloom represents the optimal window for determining fruit set rates and initiating seedless induction, while approximately 14 days post-anthesis marks the peak phase of cell division in young grape fruits. Cell numbers during this stage critically determine final fruit size [30,31]. Exogenous sprays were applied on the planned dates at full bloom (D0) and 14 days after anthesis (D14). Spraying proceeded uniformly until incipient runoff (appearance of droplets), at which point application was stopped. The concentrations and application timings employed in this study were based on established literature and preliminary optimization trials. For GA_3_, the 25 mg L^−1^ concentration aligns with commercial recommendations and previous research demonstrating optimal parthenocarpy induction in table grapes without excessive negative effects on fruit quality [5,8]. CPPU concentrations (3.0–10.0 mg L^−1^) were selected based on studies showing effective berry enlargement in ‘Summer Black’ and other cultivars, where 5–10 mg L^−1^ CPPU consistently enhanced berry size while maintaining quality parameters [5,6]. For TDZ, the 2.0 mg L^−1^ concentration was chosen as it represents approximately 40% of the CPPU concentration while accounting for TDZ’s substantially higher cytokinin activity [17]. EBR concentrations (0.2–1.0 mg L^−1^) were established based on brassinosteroid research in grapes, where concentrations below 1.0 mg L^−1^ promoted fruit development without growth inhibition [32], Setting the gradient concentration to determine the optimal concentration. The 200 mg L^−1^ SM concentration follows protocols established for parthenocarpy induction in muscadine and other grape cultivars, where this level effectively suppresses seed development [14,16]. Treatment compositions are detailed in Table 1. To characterise pre-treatment soil fertility levels, composite soil samples were collected at a 20 cm depth. Soil properties (mean ± SD) were pH 7.57 ± 0.10; electrical conductivity (EC) 0.235 ± 0.017 mS·cm^−1^; soil organic matter 49.47 ± 0.9995 g·kg^−1^; available phosphorus 20.15 ± 4.617 mg·kg^−1^; and available potassium 235.7 ± 3.1 mg·kg^−1^. The clay-sized fraction (<0.01 mm) was 345.49 ± 2.93 g·kg^−1^, and the texture was classified as medium loam according to the Kaczynski Basic Soil Texture Classification Table [33].

Fruit samples were collected weekly after treatment through the ripening period. At each sampling stage, ten berries per cluster (from three clusters per treatment) were randomly harvested, immediately frozen in liquid nitrogen, and stored at −80 °C until analysis.

### 2.3. Measurement of Fruit Quality Parameters

At each sampling date, the following quality parameters were measured for the pooled berry samples (three biological replicates). At each sampling date, the following quality parameters were measured (three biological replicates per treatment, defined as three clusters from three different vines; 10 berries per cluster; total n=30 berries per treatment per time point).

Total Soluble Solids (TSS): Berries were homogenized, and the supernatant was extracted by centrifugation (3000× *g*, 10 min). TSS was measured using a handheld refractometer at 20 °C, with values expressed as °Brix.Single berry weight and longitudinal/transverse diameters: Longitudinal and transverse diameters were measured using an electronic vernier caliper (0.01 mm precision), and single berry weight was recorded with an electronic balance (0.01 g precision).Acid Content: Fruit pulp (0.5 g) was homogenized with 20 mL distilled water, filtered, and the volume adjusted to 100 mL. A 20 mL aliquot was titrated with 0.05 M NaOH using 1% phenolphthalein as an indicator until a faint pink color lasted 30 s. TA content was calculated as:(1)$$\mathrm{TA}\left(\mathrm{mg}\cdot g^{-1}\right)=\frac{V\times N\times 0.075\times 1000}{W}$$
where *V* = titrated NaOH volume (mL), *N* = NaOH concentration (mol·L^−1^), 0.075 = conversion factor for citric acid, and *W* = sample weight (g).

### 2.4. Histological Analysis

Berry pericarp (5 mm^3^ equatorial blocks with epidermis) and pedicel segments (5 mm) were fixed in FAA (10% formalin:5% acetic acid:50% ethanol, *v*/*v*) at 4 °C for 24 h. Tissues were dehydrated through graded ethanol (50–100%), cleared in tert–butanol, embedded in paraffin, and sectioned at 8 µm. Sections were stained with Safranin O (1% in 70% ethanol) and Fast Green (0.5% in 95% ethanol) using standard protocols. Epidermal thickness of stem, cortical area of stem, cortex thickness of stem, vascular area of stem, vascular radius of stem, epidermal thickness of skin, stem area, and stem radius were quantified using ImageJ 1.54g at 100–400× magnification. Epidermal cell layers in the pericarp were counted manually at 400× magnification by identifying distinct cell boundaries perpendicular to the fruit surface. Three biological replicates with five sections per sample were analyzed. Paraffin embedding was selected over cryosectioning due to superior section quality in lignified pedicel tissues and compatibility with dual-staining protocols, while avoiding freezing artifacts common in high-water-content grape tissues.

### 2.5. Statistical Analysis

Data were recorded in Excel and analyzed using GraphPad Prism (version 9.0). Normality and variance homogeneity were tested prior to two-way ANOVA with Tukey’s post-hoc test for significance (*p* < 0.05).

## 3. Results

### 3.1. Modulatory Effects of Plant Growth Regulators on Grape Berry Development and Quality Parameters

To evaluate the effects of diverse plant growth regulators (PGRs) on berry development, T1–T11 treatments (detailed in Table 1) and a water control (CK) were applied at full bloom (D0) and 14 days post-anthesis to ‘Red Muscat of Alexandria’ and ‘Shine Muscat’. Morphological analyses revealed progressive fruit enlargement and color transition from green to yellow (‘Shine Muscat’, Figure 1) or to yellow/red hues (‘Red Muscat of Alexandria’, Appendix A), with significant morphological and size differences observed between treated groups and CK across seven time points (D0–D91).

Time-course measurements of total soluble solids (TSS) demonstrated significant treatment effects. In ‘Shine Muscat’, TSS increased consistently from 34 days after anthesis (DAA), with Treatment 9 (T9) showing significantly lower TSS than T8 (Figure 2A). Similar developmental patterns were observed in ‘Red Alexandria’ cultivar (Appendix A). Transverse rachis diameter development in ‘Shine Muscat’ was significantly modulated: Treatment 11 (25 mg/L GA_3_) maintained the largest diameter (significantly exceeding CK), while GA_3_ + CPPU (T10) significantly suppressed thickening compared to GA_3_ alone (Figure 2B). Terminal rachis diameter at D104 was significantly greater in GA_3_-treated samples (Figure 3B).

Berry weight analysis revealed cultivar-specific responses: Treatments 8 (CPPU 5.0 mg/L) and 11 (25 mg/L GA_3_) significantly increased weight in ‘Shine Muscat’ compared to CK, whereas Treatment 7 showed no significant difference (Figure 2C and Figure 3C). Total acidity (TA) dynamics differed between cultivars; ‘Shine Muscat’ initiated rapid TA decline earlier (48 DAA vs. 55 DAA in ‘Red Muscat’) and displayed significant differences between treatments (e.g., lower TA in T6 vs. T7, Figure 2D). Detailed developmental dynamics for ‘Red Muscat of Alexandria’ are presented in Appendix A. Transverse berry diameter increased temporally in both cultivars, with Treatment 5 consistently exhibiting the smallest diameter in ‘Red Muscat’. In ‘Shine Muscat’, specific EBR concentrations (T1, T2, T4) significantly reduced terminal diameter versus CK, while Treatment 7 mitigated the size-reducing effect of SM (Figure 2E and Figure 3D). Seed count analysis confirmed Treatments 6 (GA_3_ + TDZ) and 10 (GA_3_ + CPPU) achieved 100% seedlessness in ‘Shine Muscat’ (Figure 3E).

Our study reveals cultivar-specific PGR effects on grape development. In ‘Shine Muscat’, TDZ and CPPU enhanced berry size/weight while suppressing rachis thickening. GA_3_ increased weight but promoted rachis thickening and reduced TSS. EBR (≥0.2 mg/L) inhibited growth, while SM reduced berry size (mitigated by TDZ). GA_3_ + TDZ/CPPU achieved nearly 100% seedlessness, consistent with reported synergistic effects in hybrid grape cultivars [34]. ‘Red Muscat’ showed weaker responses. TDZ/CPPU demonstrate production potential, whereas GA_3_/EBR require dosage caution.

### 3.2. Effects of Plant Growth Regulators on Pedicel Microstructure Development in Grape

Different treatments have a significant impact on the physical and chemical properties and appearance of fruits. Different plant growth regulators also have different effects on the cluster stems and berry skin. With the changing market demand and the continuous optimization of consumption structure, the thickness of grape stems and berry skin has gradually become one of the key considerations for consumers when selecting high-quality grape products. However, overly thick stems affect the appearance quality of fruits and are not favored by consumers.

To assess the dynamic changes in the microstructure of the pedicel and fruit peel during berry ripening under different plant growth regulator treatments, cross-sectional paraffin sections of the pedicel and fruit peel of ‘Shine Muscat’ and ‘Red Muscat of Alexandria’ were analyzed. The developmental dynamics of pedicel structures throughout 0–104 days after anthesis (DAA) revealed distinct temporal patterns between cultivars and treatments (Figure 4). These dynamic changes included variations in epidermal thickness, cortical area, central thickness, vascular bundle development, and overall pedicel dimensions. Here, we primarily present images of paraffin sections of the pedicel and fruit skin at 104 days after anthesis (Figure 5A). Tissue morphometric assessments focused on cortical area, cortical thickness, epidermal structure, vascular tissue, and pedicel size under five plant hormone treatment conditions: EBR (T2), TDZ + GA_3_ (T6), CPPU (T8), GA_3_ (T11), and the control (CK).

Cortical area of stem (×10^3^ μm^2^) and cortex thickness of stem (μm) exhibited distinct temporal patterns between cultivars (Figure 4(ii–iii)). In ‘Red Muscat of Alexandria’, cortical area of stem and cortex thickness of stem peaked at 14 DAA before declining after 74 DAA (Figure 4A(ii,iii)). Conversely, ‘Shine Muscat’ under TDZ + GA_3_ (T6) displayed progressive cortical expansion throughout 0–104 DAA, while other treatments (EBR, CPPU, GA_3_, CK) induced reductions by maturity (Figure 4B(ii,iii)). At 104 DAA, quantitative analysis confirmed that CPPU (T8) and TDZ + GA_3_ (T6) significantly increased cortical area of stem relative to CK in ‘Shine Muscat’ (*p* < 0.05; Figure 5B(ii)), whereas GA_3_ (T11) showed no difference. Cortex thickness of stem dynamics paralleled area trends: TDZ + GA_3_ uniquely sustained cortical thickening in ‘Shine Muscat’ until 104 DAA (Figure 4B(iii)), and TDZ + GA_3_/CPPU enhanced cortex thickness of stem versus CK (Figure 5B(ii)). Notably, T6-treated ‘Shine Muscat’ exhibited significantly greater cortex thickness of stem than ‘Red Muscat of Alexandria’ at maturity (Figure 5B(iii)).

Epidermal thickness of stem increased continuously in ‘Red Muscat of Alexandria’ throughout 0–104 DAA across all treatments (Figure 4A(i,vi)). ‘Shine Muscat’ showed similar thickening until 74 DAA but declined at 104 DAA (Figure 4B(i,vi)). Treatment-specific effects emerged at maturity: CPPU (T8) significantly enhanced epidermal thickness of stem in ‘Red Muscat of Alexandria’ versus CK (*p* < 0.05; Figure 5B(vi)), while TDZ (T6) increased it in ‘Shine Muscat’ (Figure 5B(vi)). Microscopy revealed a single epidermal cell layer surrounding the cortex, with the skin epidermis comprising 2–3 cell layers covered by a cuticle (Figure 5A).

Vascular area of stem and vascular radius of stem fluctuated markedly during development (Figure 4(iv,v)). At maturity, GA_3_ (T11, 25 mg/L) significantly increased vascular area of stem relative to CK in ‘Shine Muscat’ (Figure 5(v)), supporting gibberellin’s role in vascular thickening. CK-treated ‘Shine Muscat’ displayed higher vascular pole ratios than RMA, and T11 induced a notable recovery in this ratio specifically in ‘Shine Muscat’ (Figure 5(iv)). Conversely, T6 and T8 treatments reduced vascular area of stem in ‘Shine Muscat’ compared to RMA (Figure 5A).

Stem radius (viii) and stem area (vii) varied temporally: RMA stem radius increased from D0, plateaued at D14, and decreased by D104, while ‘Shine Muscat’ stem radius increased from D0 and ceased growth by D24 (Figure 4(viii)). Stem area trends mirrored radius dynamics (Figure 4(vii)). At 104 DAA, TDZ (T6) significantly increased stem area and stem radius in ‘Shine Muscat’ versus CK (Figure 5(vii,viii)), whereas GA_3_ (T11) showed no difference. CK ‘Shine Muscat’ had larger stem area than RMA (Figure 5(vii)). Scale bar measurements confirmed ‘Shine Muscat’ pedicels were generally larger with enlarged cells (Figure 5A).

Concurrent with pedicel changes, pericarp development showed treatment-modulated effects. Initially thin exocarps expanded into robust multilayered structures by maturity, accumulating pigments and metabolites (Figure 5A). CPPU significantly enhanced epidermal thickness of skin in RMA (Figure 5B(vi)), while TDZ increased it in ‘Shine Muscat’ (Figure 5B(vi)). Epidermal cell layer analysis indicated ‘Shine Muscat’ formed two layers earlier (14 DAA vs. 24 DAA in RMA) and had more layers in CK at 104 DAA. CPPU increased epidermal layers in RMA, whereas EBR (T2) reduced layers in both cultivars (Table 2).

## 4. Discussion

### 4.1. Cultivar-Specific Responses to PGRs and Implications for Seedless Production

The modulatory effects of PGRs on berry development varied markedly between ‘Shine Muscat’ and ‘Red Muscat of Alexandria’, highlighting the influence of genetic background. For instance, GA_3_ + TDZ or GA_3_ + CPPU achieved 100% seedlessness in ‘Shine Muscat’ (Figure 3E), aligning with the high efficacy of GA_3_-streptomycin (SM) combinations in grapes [35]. Based on previous studies, we hypothesize that SM may disrupt ovule development by inhibiting normal cell division, while GA_3_ potentially affects embryogenesis [36], suggesting a promising foundation for commercial seedless protocols that requires molecular validation. However, SM also reduced berry weight and TSS (Figure 2C and Figure 3C), likely due to impaired seed-derived growth signals or metabolic shifts, as observed in ‘Red Alexandria’, where SM-treated berries exhibited smaller transverse diameters (Appendix A).

Conversely, TDZ and CPPU consistently promoted berry enlargement and weight gain in ‘Shine Muscat’ (Figure 2C and Figure 3C), presumably by acting as potent cytokinins that stimulate cell division [37,38], aligning with reports that have shown that TDZ increases fruit size and sugar content in similar grape varieties [11]. Yet, both suppressed rachis thickening (Figure 2B), a common trait among cytokinin analogs that we hypothesize may stem from altered auxin–ethylene signaling [39,40], though direct molecular evidence is needed to confirm this mechanism. This suppression, while beneficial for reducing undesirable pedicel coarseness, could compromise structural integrity during postharvest handling [39,41]. GA_3_, despite increasing berry weight, exacerbated rachis thickening and reduced TSS (Figure 2A,B), indicating a need for dosage optimization to balance yield and quality [42,43]. EBR’s inhibitory effects on transverse berry diameter at concentrations ≥ 0.2 mg/L (Figure 2E) aligning with the role in ‘Ruidu Hongyu’ grapes [11].

The weaker PGR responses in ‘Red Muscat of Alexandria’ (such as minimal effects on dry weight or TSS) suggest inherent physiological buffers or lower hormone receptivity. We propose that this weaker response may be caused by differences between species, as ‘Shine Muscat’ is a hybrid of *Vitis labruscana* × *Vitis vinifera* L., while ‘Red Muscat of Alexandria’ is a variety from *Vitis vinifera.* L. Based on previous research, we hypothesize that the differential responses of *Vitis vinifera* L. varieties in terms of auxin–gibberellin synergistic pathways and sugar transport capacity may explain their varying responsiveness to growth-promoting regulators such as GA_3_. Similarly, we suggest that hybrids of *Vitis labruscana* × *Vitis vinifera* L. may show different sensitivity to ABA-type regulators due to their potentially distinct promoter structures and stress response gene expression patterns [19,20,21,22,23], though detailed molecular characterization is needed to confirm these hypotheses.

### 4.2. Microstructural Modifications and Marketability Implications

PGR treatments profoundly influenced pedicel and pericarp microstructure, with potential consequences for fruit durability and consumer appeal. TDZ and CPPU increased cortical area and epidermal thickness in both cultivars (Figure 4(ii,iii) and Figure 5B(vi)), potentially enhancing pericarp robustness. We hypothesize that this structural modification may correlate with improved disease resistance, as thicker epidermal layers could potentially act as barriers against pathogen invasion [44,45], though controlled pathogen challenge experiments would be needed to validate this assumption.

The observed microstructural changes have significant implications for postharvest performance and marketability. Pericarp thickness and cellular density directly influence water loss rates and pathogen resistance during storage, as the fruit skin serves as the primary protective barrier against moisture evaporation and microbial invasion [46]. Enhanced epidermal thickness, as observed with CPPU and TDZ treatments, may potentially extend storage life by reducing transpiration rates and maintaining fruit turgor. Furthermore, the integrity of the waxy cuticle layer and cell wall composition, particularly lignin and pectin content and cross-linking degree, determine berry firmness and resistance to mechanical damage during handling and transport [46].

From a consumer preference perspective, the visual appeal of grape berries is largely dependent on skin color uniformity, surface integrity, and gloss, which are directly influenced by pericarp microstructure [47]. The accumulation of phenolic compounds such as proanthocyanidins in the pericarp not only contributes to color development but may also enhance antioxidant capacity, potentially extending shelf life through free radical scavenging [46]. For example, CPPU increased epidermal cell layers in ‘Red Muscat of Alexandria’, while ‘Shine Muscat’ exhibited earlier and greater layer accumulation (Table 2), potentially explaining its superior field resilience and enhanced marketability characteristics.

TDZ also induced pedicel thickening through cortical expansion (Figure 4B(ii,iii) and Figure 5B(ii)) [48,49], which, while supporting larger berry weights (Figure 3C), may detract from aesthetic quality as consumers prefer slender pedicels. This trade-off underscores the importance of targeted applications—e.g., restricting TDZ to early development to mitigate coarseness [50]. GA_3_ increased vascular bundle area (Figure 5B(v)), consistent with its reported role in promoting vascular tissue growth [51], but had no significant effect on overall pedicel size, suggesting compartmentalized impacts. EBR reduced epidermal cell layers (Table 2), which we hypothesize may potentially weaken disease defense [52], a vulnerability that could require mitigation in humid climates [53], though direct disease susceptibility testing would be necessary to confirm this relationship.

### 4.3. Research Method Limitationss

This study advances understanding of PGR interactions but presents several methodological constraints requiring future investigation. At the molecular level, the mechanisms underlying EBR–cytokinin interactions or TDZ-induced rachis suppression remain speculative; transcriptomic or proteomic analyses could elucidate these pathways [54]. Chen et al. demonstrated that the FvMYB10 mutation causes strawberry color loss by inhibiting anthocyanin synthesis genes [55], while EBR treatment has been shown to modulate plant growth and stress responses through brassinosteroid signaling pathways in grapes [56]. Vrobel et al. established a multi-hormone HILIC-MS/MS analysis technique that can quantitatively reveal hormone interaction networks [54].

Environmental control limitations also merit consideration. Despite greenhouse cultivation aimed at environmental control, potential microclimatic variations within the facility were not systematically monitored or accounted for. Light gradients, temperature fluctuations, and humidity variations can occur even in controlled environments and may confound treatment effects. The random allocation of clusters across greenhouse locations partially mitigated this concern, but future studies should incorporate comprehensive environmental monitoring (light sensors, temperature loggers) at individual plant locations to better account for spatial heterogeneity. Additionally, while environmental variables (e.g., light, temperature) were controlled, they were not systematically varied, yet field studies show they modulate PGR efficacy (e.g., Quamruzzaman et al. confirmed that the effect of PGR under salt stress varies depending on environmental fluctuations [57])—future work should integrate controlled climate factor manipulation.

Postharvest evaluation represents another research gap. Our study focused primarily on harvest-time quality parameters without evaluating postharvest performance or conducting controlled storage trials. While we observed enhanced pericarp thickness that may theoretically improve storage life, direct validation through postharvest studies measuring water loss rates, firmness retention, and pathogen resistance under various storage conditions remains necessary [46,47]. Additionally, consumer sensory evaluation linking observed microstructural changes to actual preference data would strengthen the marketability implications.

### 4.4. Risk Assessment and Regulatory Challenges

Beyond methodological constraints, risk management is central to responsible deployment of PGRs. Long-term environmental risks remain insufficiently characterized: the migration and transformation behavior of PGRs in soil may threaten ecological safety [58], and synthetic PGRs may exert persistent physiological effects in agricultural systems [59]. These concerns motivate formal environmental risk assessments alongside efficacy trials.

At the regulatory level, PGR usage in grape production varies significantly across regions, reflecting differences in regulatory frameworks, market demands, and production practices [60]. While countries like the United States, Australia, and Chile have established comprehensive registration systems for PGRs with defined maximum residue limits, many developing regions operate under less stringent oversight, potentially leading to inconsistent application and varying safety standards [15]. This regulatory heterogeneity creates challenges for international trade, as export markets increasingly demand compliance with the most restrictive standards. The lack of harmonized global regulations particularly affects substances like streptomycin, where some jurisdictions permit limited use while others maintain complete prohibitions, creating confusion among producers and potential trade barriers [61].

The use of streptomycin (SM) in our experimental design exemplifies these regulatory and public health complexities. While SM demonstrated parthenocarpic potential, its classification as an antibiotic presents multiple risks: (1) residue accumulation in edible tissues may pose direct health risks to consumers, (2) environmental release promotes the development and dissemination of antibiotic resistance genes (ARGs) in soil and plant-associated microorganisms, potentially contributing to the global antibiotic resistance crisis, and (3) most jurisdictions, including the European Union, strictly prohibit or severely restrict antibiotic use in food crop production due to these public health concerns [15]. The default maximum residue limit (MRL) of 0.01 mg/kg is practically unachievable for agricultural applications [16]. Therefore, while our SM results provide valuable insights into alternative parthenocarpy mechanisms, commercial implementation would require extensive toxicological evaluation, residue studies, and regulatory approval.

### 4.5. Market and Application Prospects

From a consumer and market perspective, seedless table grapes command premium prices in major markets including North America, Europe, and Asia [60,62], and consumer preference studies consistently demonstrate that seedlessness ranks as a primary purchasing criterion, often outweighing size or sugar content [63]. This market demand has driven widespread adoption of parthenocarpy-inducing treatments, particularly GA_3_-based protocols, across major grape-producing regions [60]. However, consumer awareness of PGR use remains limited; increasing transparency regarding production methods and clear communication of safety assessments may influence future market acceptance. The growing demand for organic and minimally processed foods presents challenges for PGR-dependent production systems, necessitating development of alternative approaches such as genetic selection for natural parthenocarpy or reduced-input management strategies [63].

Economically, PGR usage presents a complex cost–benefit profile that varies by production scale, target market, and regulatory environment. While PGR applications typically increase direct production costs through purchase, application, and compliance expenses, premium prices achieved for high-quality seedless grapes can justify these investments [60]. Smaller producers may face disproportionate compliance costs, particularly for residue testing and documentation. In addition, the risk of harvest losses due to improper timing or dosage emphasizes the need for producer education and technical support [64]. Long-term sustainability requires balancing immediate production gains against potential costs associated with environmental remediation, consumer backlash, or regulatory restrictions, suggesting that integrated approaches combining PGRs with sustainable practices may offer the most viable economic pathway [64].

From an applied perspective, optimizing combinatorial sprays (e.g., EBR + cytokinins for enhanced fruit quality; GA_3_ + CPPU for seedlessness) could improve both quality and yield, but dosage standardization across cultivars and environments is essential to avoid negatives like vitamin C loss or oversized pedicels. Additionally, exploring natural hormone analogs or gene-editing approaches may offer sustainable alternatives to synthetic PGRs.

## 5. Conclusions

This study establishes that PGRs are effective, cultivar-specific levers for steering berry development and quality. Rather than prescribing a single recipe, our results support a decision framework that: (i) applies genotype- and stage-appropriate dosing; (ii) uses microstructural readouts (cortical and epidermal metrics, vascular traits) as early indicators of downstream quality; and (iii) explicitly balances enlargement with sensory, postharvest, and sustainability outcomes. Embedding these principles can reduce empirical trial-and-error and enable reproducible, precision viticulture across environments.

Future work should translate these principles into predictive, field-ready tools by integrating multi-season, multi-site datasets; linking histology to non-destructive sensing for on-vine monitoring; quantifying postharvest and flavor performance; and assessing environmental and regulatory sustainability of PGR portfolios. These advances will move PGR use from heuristic practice to knowledge-driven, sustainable management in commercial grape production [61].

## Figures and Tables

**Figure 1 biology-14-01270-f001:**
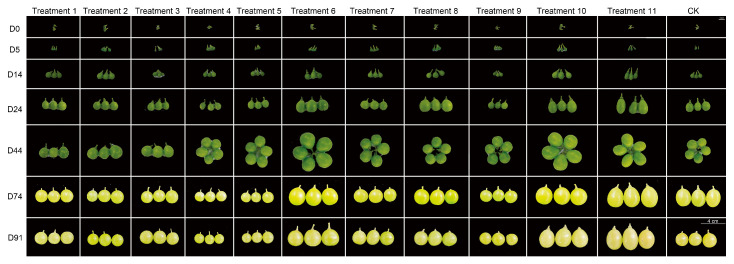
Phenotypic development of *V. vinifera* ‘Shine Muscat’ grape berries under various plant growth regulator (PGR) and streptomycin (SM) treatments. Berry clusters were photographed at multiple time points from day 0 (full bloom) to day 91 after anthesis (DAA). Treatments 1–11 correspond to the different PGR/SM combinations as detailed in Table 1, with CK representing the water-treated control. The scale bar represents 4 cm.

**Figure 2 biology-14-01270-f002:**
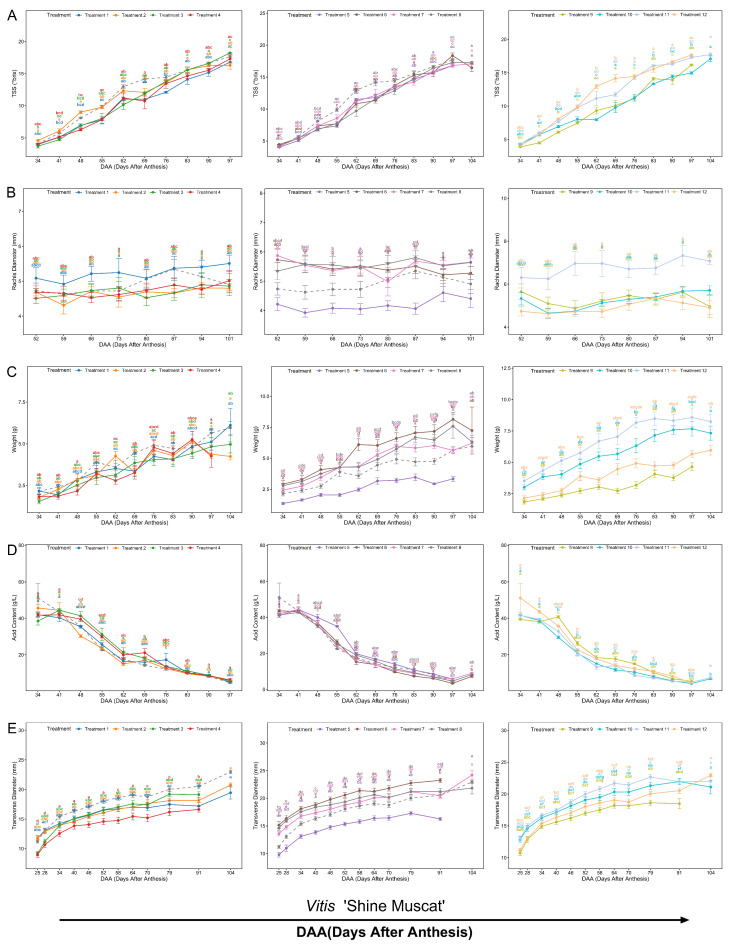
Developmental dynamics of ‘Shine Muscat’ grape berries under plant growth regulator (PGR) and streptomycin (SM) treatments. (**A**) Total soluble solids (TSS) content. (**B**) Rachis diameter. (**C**) Berry weight. (**D**) Acid Content (TA). (**E**) Transverse diameter. Treatment conditions 1–11 correspond to the PGR/SM combinations detailed in Table 1, with CK (Treatment 12) as the water-treated control. Lowercase letters indicate significant differences among treatments at the same time point (*p* < 0.05).

**Figure 3 biology-14-01270-f003:**
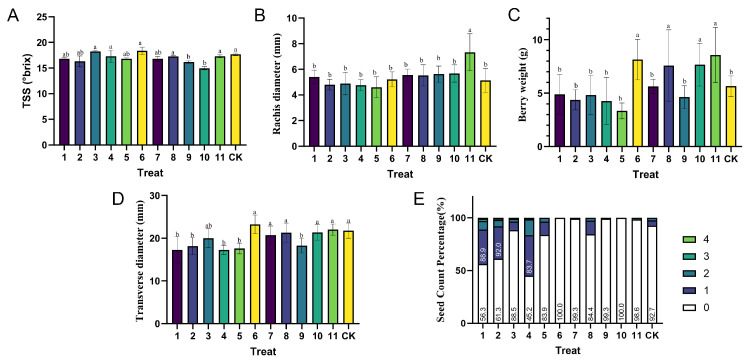
Terminal developmental parameters of ‘Shine Muscat’ grapes under plant growth regulator treatments at D104. Panels A-E quantify treatment effects on quality and ampelometric characteristics: (**A**) total soluble solids (TSS, °Brix). (**B**) Transverse Rachis diameter (mm). (**C**) Berry fresh weight (g). (**D**) Transverse berry diameter (mm). (**E**) Seed count percentage (0-seed category %). Lowercase letters indicate significant differences among treatments (*p* < 0.05).

**Figure 4 biology-14-01270-f004:**
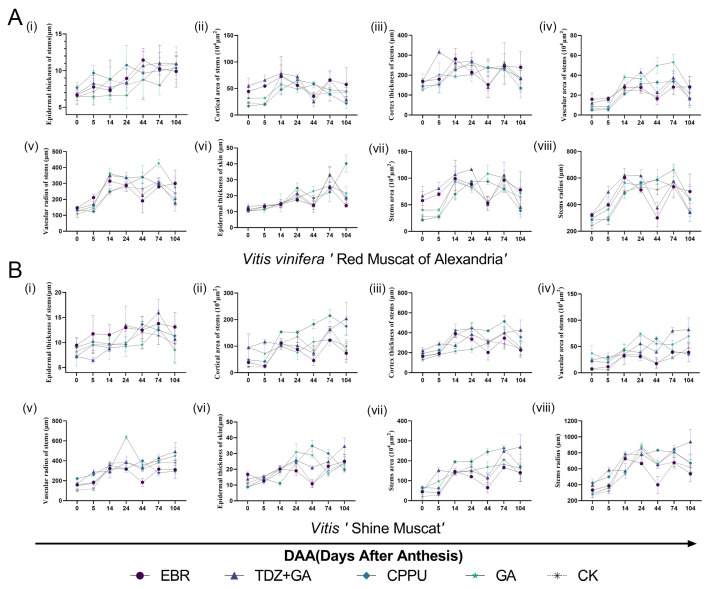
Developmental dynamics of pedicel structures in ‘Red Muscat of Alexandria’ (**A**) and ‘Shine Muscat’ (**B**) under plant growth regulator treatments during 0–104 days after anthesis (DAA). Parameters measured were as follows: (**i**) Epidermal thickness (μm), (**ii**) cortical area (×10^3^ μm^2^), (**iii**) central thickness (μm), (**iv**) vascular bundle area (×10^3^ μm^2^), (**v**) vascular bundle/pedicel ratio (%), (**vi**) epidermis/pedicel ratio (%), (**vii**) pedicel area (×10^3^ μm^2^), (**viii**) pedicel radius (μm).

**Figure 5 biology-14-01270-f005:**
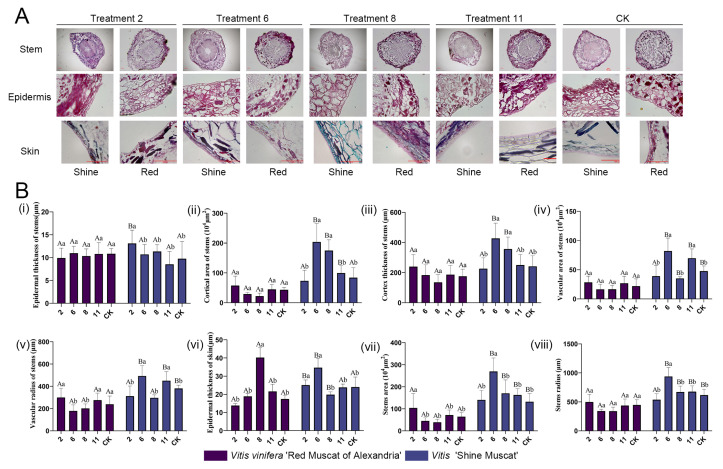
Histomorphometric analysis of pedicel structures in ‘Shine Muscat’ and ‘Red Muscat of Alexandria’ under selected treatments at 104 DAA. (**A**) Transverse sections depicting epidermis, cortex, and vascular tissues at maturity (104 DAA; scale bars = 100 μm). (**B**) Quantitative comparison of (**i**) epidermal thickness (μm), (**ii**) cortical area (×10^3^ μm^2^), (**iii**) vascular bundle area ratio (%), (**iv**) vascular pole ratio (%), (**v**) epidermal cell radius (μm), (**vi**) stem area (×10^3^ μm^2^). Data: mean ± SD; lowercase letters indicate significant differences between different treatments within a cultivar, while uppercase letters indicate significant differences between different varieties under the same treatment (*p* < 0.05). Treatments 2, 6, 7, 8, 11, and CK detailed in Table 1, with purple and blue denoting ‘Red Muscat’ and ‘Shine Muscat’, respectively.

**Table 1 biology-14-01270-t001:** Exogenous spray treatment combinations for ‘Shine Muscat’ and ‘Red Alexandria’ grapes at full bloom and 14 Days post-anthesis.

Treatment Group	Full Bloom (D0)	14 Days Post-Anthesis (D14)
1	EBR 0.2 mg/L	EBR 0.5 mg/L
2	EBR 0.5 mg/L	EBR 0.5 mg/L
3	EBR 0.8 mg/L	EBR 0.5 mg/L
4	EBR 1.0 mg/L	EBR 0.5 mg/L
5	0.5 mg/L EBR + 200 mg/L SM	EBR 0.5 mg/L
6	2 mg/L TDZ + 25 mg/L GA_3_	TDZ 2 mg/L
7	200 mg/L SM + 2 mg/L TDZ	TDZ 2 mg/L
8	CPPU 5.0 mg/L	CPPU 10 mg/L
9	200 mg/L SM + CPPU 5.0 mg/L	CPPU 10 mg/L
10	25 mg/L GA_3_ + CPPU 3.0 mg/L	25 mg/L GA_3_ + CPPU 5.0 mg/L
11	25 mg/L GA_3_	25 mg/L GA_3_
CK	Solvent (Water)	Solvent (Water)

Note: EBR: 2,4-Epibrassinolide Solution, SM: Streptomycin, GA_3_: Gibberellic Acid, TDZ: Thidiazuron, CPPU: Forchlorfenuron.

**Table 2 biology-14-01270-t002:** Number of epidermal cell layers in the pericarp of ‘Red Muscat of Alexandria’ and ‘Shine Muscat’ under plant growth regulator treatments during berry development.

Treatment	Cultivar	Days After Anthesis (DAA)						
		D0	D5	D14	D24	D44	D74	D104
T2	Red	1	1	1	2	2	2	1
	Shine	1	1	2	2	2	1	2
T6	Red	1	1	1	2	2	3	2
	Shine	1	1	2	3	3	3	3
T8	Red	1	1	1	2	2	3	3
	Shine	1	1	1	2	2	3	3
T11	Red	1	1	2	2	2	3	3
	Shine	1	1	2	3	2	3	2
CK	Red	1	1	1	2	1	3	2
	Shine	1	1	2	2	1	3	3

Note: Values represent the modal number of epidermal cell layers observed in cross-sections.

## Data Availability

The data presented in this study are available on request from the corresponding author.

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
