# Peer review of "Synergistic Effects of Gibberellic Acid, Forchlorfenuron, Thidiazuron, and Brassinosteroid Combinations on Seedless Berry Development and Quality Enhancement in ‘Shine Muscat’ and ‘Red Muscat of Alexandria’ Grapes"

_biology, 2025, doi:10.3390/biology14091270_

Round 1
Reviewer 1 Report
Comments and Suggestions for Authors
The paper is well thought out. The experiment is complex and the results are clearly presented. Additional corrections are needed to the Material and methods section. The sction Conclusion needs to be rewritten with a clear presentation of the conclusions based on the results obtained. It is also necessary to emphasise in the paper the limited use of PGRs in accordance with regulatory requirements, the importance of using PGRs as adjuvants in regular grape production and the possible effects on humans health as grape users. These issues are particularly important considering that in most countries the use of streptomycin is prohibited except for scientific purposes.
After the corrections have been made, I propose the paper for publication.

Reviewer 2 Report
Comments and Suggestions for Authors
Dear Authors,
The manuscript investigates the impact of Gibberellic Acid, CPPU, Thidiazuron, and Brassinosteroid Combinations on Seedless Berry Development and Quality Enhancement in ’Shine Muscat’ and ’Red Muscat of Alexandria’ Grapes. The authors do not indicate if the study spans one, two, or more years and includes a broad spectrum of morphological and biochemical measurements. Although the study addresses a relevant topic in sustainable viticulture and includes detailed datasets, several critical concerns significantly limit the scientific robustness, clarity, and generalizability of the manuscript. I therefore recommend that the manuscript needs some specific revisions before publication.
The introduction presents a general background, identifies knowledge gaps, and formulates a hypothesis. Authors must clearly articulate the scientific hypothesis and specify whether the applications of gibberellic acid, CPPU, thidiazuron, and brassinosteroid combinations are expected to have additive or synergistic effects. Otherwise, the justification for the factorial design remains vague.
Materials and Methods
The description of the field site, plant material, and design lacks critical reproducibility details. Indicate soil fertility parameters before treatment. Also, very important are climatic conditions for specific years of study – they strongly influence the results.
Lines 113-114: The following sentence: ”For ‘Shine Muscat’, 20 vines were cultivated under the same management as ‘Muscat Hamburg’. ” – needs to be complete or rephrased to be understood..
The frequency and method of foliar application are stated, but no justification is given for the dose or timing. Provide literature-based or preliminary data-based justification for the selected concentrations and application intervals. Without this, the choice appears arbitrary.
The methodology for physiological and biochemical quality parameters is sound in general.
The study has two varieties (’Shine Muscat’ and ’Red Muscat of Alexandria’), but there are data exposed only for Shine Muscat in Figure 1. Phenotypic development of V. vinifera ’Shine Muscat’ grape berries under various plant growth regulator (PGR) and streptomycin (SM) treatments.
Conclusion
Although the manuscript presents an extensive dataset on gibberellic acid, CPPU, thidiazuron, and brassinosteroid combinations on seedless berry development and quality enhancement in ’Shine Muscat’ and ’Red Muscat of Alexandria’ grapes, the study suffers from several issues of scientific rigor, transparency, and focus. Therefore, despite the apparent volume of data, the authors need to improve the materials and methods section. A significant overhaul would be required, including a reformulation of the study design, statistical validation, and language refinement.
Thank you!

Reviewer 3 Report
Comments and Suggestions for Authors
The authors investigated the effects of five plant growth regulators (GA₃, EBR, CPPU, TDZ, and streptomycin) and their combinations on berry development, seedlessness, quality parameters, and microstructural changes in two grape cultivars (‘Shine Muscat’ and ‘Red Muscat of Alexandria’). They found cultivar-specific responses, with GA₃+TDZ and GA₃+CPPU achieving nearly complete seedlessness in ‘Shine Muscat’, and TDZ/CPPU promoting berry enlargement and pericarp robustness, whereas GA₃ and EBR had variable or adverse impacts.
While the study presents valuable data on PGR interactions in grapes, the manuscript in its current form is too lengthy, contains unnecessary methodological and descriptive detail, and occasionally draws conclusions beyond the presented evidence. I recommend major revision with the following priorities.
L34–45: The introduction effectively establishes the importance of PGRs in viticulture but is overly lengthy and occasionally unfocused. Condense the background to highlight the research gap more clearly, and ensure all statements (e.g., “remain poorly understood”) are supported by appropriate references.
L86–105: The description of cultivar characteristics is partly repetitive and blends into methods. Move cultivar-specific traits (e.g., “thin skin, high sugar”) to the discussion or keep them concise here. Clarify whether the stated “identical field management” applies to both cultivars in greenhouse conditions.
L118–125: Experimental design is adequately described, but the replication scheme is ambiguous. Explicitly state the total number of biological replicates per treatment and how randomization was applied to minimize environmental bias.
L143–159: The histological analysis section is technically detailed, but some steps (e.g., ethanol gradient times, staining durations) could be shortened for clarity. Ensure all parameters are necessary for reproducibility. Include justification for paraffin embedding versus other sectioning methods.
L174–200: Seven time points are reported for berry development, yet the utility of presenting the full time-course is questionable for the main conclusions. Consider focusing on key developmental stages (e.g., early fruit set, véraison, maturity) to simplify results presentation without losing biological meaning.
L217–223: Subheading inconsistencies between methods and results (e.g., “Cortical area” in results vs. “cortex thickness” in methods) reduce clarity. Align terminology across sections.
L250–256: The pericarp development description could be supported by quantitative epidermal cell layer data directly in the main text instead of relegating to supplementary materials.
L258–296: The discussion correctly emphasizes cultivar-specific PGR responses but overstates mechanistic conclusions without molecular evidence. Avoid definitive causal language (“providing a robust foundation”) unless supported by direct mechanistic assays. Instead, present as hypotheses requiring further validation.
L297–314: The microstructural implications for marketability are interesting but need stronger linkage to postharvest performance or consumer preference data. I recommend including citations from postharvest studies linking pericarp/cortex traits to storage life and visual appeal.
L315–335: Limitations are acknowledged, but some critical aspects are missing. For example, the potential variability in environmental conditions even within a greenhouse (light gradients, temperature fluctuations) is not addressed. In addition, the potential health and regulatory concerns related to streptomycin use in food crops warrant more thorough discussion.
L336–343: The conclusions accurately summarize key findings but could be more concise. Avoid repeating results; instead, focus on the broader implications for precision viticulture and future research directions.
Also, condense the introduction, methods, and results to focus on essential findings.
Clarify experimental replication, randomization, and terminology consistency.
Streamline data presentation by reducing redundant time points and integrating supplementary data where critical.
Moderate mechanistic claims unless supported by molecular data; frame as hypotheses where appropriate.
Enhance discussion of environmental, postharvest, and regulatory implications, particularly regarding streptomycin use.
Following thorough revision and clarification of these points, I will be able to make a final decision on the manuscript.
Round 2
Reviewer 3 Report
Comments and Suggestions for Authors
The authors have carefully and satisfactorily addressed all of my previous comments. The manuscript has been substantially improved in clarity, scientific rigor, and overall presentation. I am satisfied with the revisions and consider the paper suitable for publication. I therefore recommend acceptance in its current form.